# What Is on the Menu?—A Quantitative Analysis on Label Format among (Potential) Restaurant Guests and Restaurant Owners

**DOI:** 10.3390/ijerph182312500

**Published:** 2021-11-27

**Authors:** Nadja S. J. Hanssen, Joost O. Linschooten, J. Hein M. van Lieverloo, Annet J. C. Roodenburg

**Affiliations:** Department Food & Industry, HAS University of Applied Sciences, 5223 DE ‘s-Hertogenbosch, The Netherlands; nadja.hanssen@outlook.com (N.S.J.H.); j.linschooten@has.nl (J.O.L.); h.vanlieverloo@has.nl (J.H.M.v.L.)

**Keywords:** food service, out-of-home eating, restaurant guests, restaurant owners, menu labeling, health logo, calorie information

## Abstract

About 20% of energy intake in the Netherlands is consumed out-of-home. Eating out-of-home is associated with higher energy intake and poorer nutrition. Menu labeling can be considered a promising instrument to improve dietary choices in the out-of-home sector. Effectiveness depends on the presentation format of the label and its attractiveness and usability to restaurant guests and restaurant owners. This exploratory study investigated which menu labeling format would be mostly appreciated by (a) (potential) restaurant guests (n386) and (b) the uninvestigated group of restaurant owners (n41) if menu labeling would be implemented in Dutch full-service restaurants. A cross-sectional survey design was used to investigate three distinct menu labeling formats: a simple health logo; (star) ranking and calorie information. Questionnaires were used as study tool. Ranking has been shown to be the most appreciated menu labeling format by both (potential) restaurant guests and owners. Statistical analysis showed that label preference of potential restaurant guests was significantly associated with age, possibly associated with level of education, and not associated with health consciousness. In summary, we found that ranking is the most appreciated menu label format according to both (potential) restaurant guests and restaurant owners, suggesting it to be a promising way to improve healthy eating out-of-home.

## 1. Introduction

The prevalence of overweight and obesity in the world is rising [1]. The Netherlands is one of the countries in which the number of people suffering from being overweight is increasing at an alarming rate: In 2020, about 50% of Dutch adults were overweight or obese [2]. Given that overweight and obesity increase the risk of noncommunicable diseases, there is a clear imperative for action to tackle this problem [1]. As per the most recent numbers, 20% of the calorie intake of Dutch people is consumed out-of-home [3], e.g., in (quick service) restaurants, bars, hotels and workplaces, but also on-the-go (in public transport, vending machines, etc.). Since eating out-of-home is associated with higher energy intake and poorer nutrition [4,5] restaurants can form part of the solution to this worrying trend [6,7]. Hence, menu labeling, referring to the provision of information on calories and nutrients on restaurant menus, can be used as an instrument to improve dietary choices [6,8], both by helping consumers make the healthy choice and by stimulating food service professionals to reformulate menu items and reduce portion sizes [9].

Since the introduction of menu labeling in the out-of-home-sector, an increasing amount of research has focused on its effectiveness. Most studies examined the effect of menu labeling on consumer behavior. Only limited scholarly effort has been made to examine the effect of menu labeling on restaurants’ behavior. This makes it difficult to draw conclusions about the effect of menu labeling on reformulation of menu items. However, preliminary evidence suggests a positive effect of menu labeling on the calorie content of menu items [10]. A more recent study from Bleich et al. [11] in large chain restaurants does also show a decline in calories in menu items introduced between 2013 and 2018, but no improvement of the macronutrient composition. A systematic review performed by Rincón-Gallardo Patiño et al. [9] evaluated the effect of mandatory and voluntary menu labeling policies in various countries on reformulation of menu items or reduction in portion sizes. The authors found reductions in energy for newly introduced menu items only with a mandatory policy in the United States. Regarding the effect of menu labeling on calories ordered or consumed research results are inconsistent. For example, a meta-analyses performed by Littlewood et al. [12] including 12 studies conducted both in real-life and in experimental settings demonstrated menu labeling to be effective in reducing the number of calories ordered and consumed in both settings. Cecchini and Warin [13] suggest that menu labeling assists people in making a healthy choice, but has a less convincing effect on calories ordered or consumed due to a widely varying individual response to the introduction of menu labeling. Several other studies have even reported no effect of menu labeling on either calories or nutrients ordered and consumed [14,15]. This inconsistency is most probably due to the wide variety of factors that are involved in the outcomes, such as restaurant settings, type of consumers and the different menu labeling formats used [8,14,16,17]. A systematic review performed by Bleich et al. [10] also concluded that the effectiveness of menu labeling on calories ordered and consumed may differ between types of restaurants. This leaves the food service sector in a difficult position, without knowing about the reasons for menu labeling formats’ effectiveness, any measures might remain ultimately ineffective in improving dietary choices with persisting health risks associated with overweight and obesity.

Calorie labeling in fast-food restaurants became mandatory in New York City in 2008 [18]. Since then, a lot of research has been undertaken on multiple aspects with regard to calorie labeling and its efficacy [15]. Yet, the extant literature lacks knowledge about effects of menu labeling formats other than calorie labeling [8]. A systematic review performed by Fernandes et al. [8] suggests that, in cafeterias, labeling with qualitative information may improve dietary choices to a greater extent than calorie labeling. The effectiveness of this qualitative information also seems to be determined by its presentation format, such as healthy food symbols (e.g., keyhole) or a traffic light system [8]. Since there is no legislation on menu labeling in the Netherlands, so far only limited scholarly effort has been made to explain the effect of menu labeling on calorie intake of Dutch consumers. A Dutch study performed by Hoefkens et al. [19] among guests of university canteens reported the liking and attractiveness of the presentation formats of nutrition information to be important determinants for the use of menu labels by guests of university canteens. A study performed by Vyth et al. [20] among worksite cafeteria managers examined which factors are important in determining the willingness of worksite cafeteria managers to implement a health logo. The health logo used in this study was the Choices logo. Results showed that success factors for implementation of the Choices logo by catering managers are consistency of the logo with the beliefs of the manager about healthy food, ability of the manager to observe and communicate the results of logo implementation and limited workload to implement the logo [20]. Little is known about the appreciation by guests and food service professionals of the different menu labeling formats in the food service setting.

The aim of this exploratory research is therefore to examine which menu labeling formats would be most appreciated by (potential) restaurant guests and by restaurant owners if menu labeling were to be implemented in full-service restaurants in the Netherlands.

## 2. Materials and Methods

### 2.1. Data Collection 

Data was collected in the period April 2014 to May 2014. The study population consisted of both (potential) guests and owners of full-service restaurants. To gain insight into the preference for a certain menu labeling format, both (potential) restaurant guests as well as restaurant owners were asked which menu labeling format they would prefer, based on a fictive menu which was presented in a questionnaire.

(Potential) restaurant guests: Participants among (potential) restaurant guests were recruited online using fora, Twitter, and Facebook. Because of this sampling design selection bias causes non-representativeness of the sample. Furthermore, since the recruitment was done by students from HAS university of applied sciences, relatively young and higher educated (potential) restaurant guests are more likely to be selected and are overrepresented in the study population. The questionnaire was distributed through social media. The participants were asked to fill in the questionnaire by clicking on a link to Thesistools.com. Participants were informed about the study at the first page of the questionnaire. Participation was anonymous, the answers could not be traced back to the participants. With completing the questionnaire participants gave their informed consent. Only participants who live in the Netherlands or Belgium were eligible. (Potential) restaurant guests were excluded from participation if they live elsewhere or if they could not read Dutch.

Restaurant owners: Participants among restaurant owners were recruited through a call in catering trade magazines and local newspapers (‘s-Hertogenbosch) and through personal approach by the researchers. Restaurant owners were excluded from participation if they had no menu or a daily changing menu and if they worked with a take-away concept without a place to sit. Restaurants that indicated their willingness to participate in the study and met the inclusion criteria received an information letter after which the questionnaire was handed to the restaurant owners by the researchers at an agreed date and time. During this visit restaurant owners had the opportunity to ask remaining questions and were asked face-to-face to fill out the questionnaire on paper. The questionnaire was picked up a week later, providing the participant with enough time to decide about participation in the study. With completing the questionnaire participants gave their informed consent. All answers were anonymized before analysis.

### 2.2. Research Instrument

The questionnaire presented to the (potential) restaurant guests (Appendix A) was tested for comprehensibility with six participants. Participants saw fictive menus with all three different formats incorporated, but the order in which these formats were presented was randomized. The content of all the menus was the same, except for the incorporated menu labeling format. The menu labels were assigned to the menu items based on the criteria for main courses as composed by the Choices foundation [21]. The questionnaire was developed based on Rogers’ Diffusions of Innovation Theory [22]. In addition to questions about the different menu labeling formats, the (potential) restaurant guests were also presented with questions on health consciousness to examine whether the preference for a certain menu labeling format is associated with their own health consciousness [23]. The questions on health consciousness were formulated based on the theory of Dutta-Bergman (2004) [24]. Health consciousness was determined based on 5 questions using a 5-point Likert scale (1 = strongly disagree, 5 = strongly agree). Outcomes on health consciousness were categorized in three groups, based on the Likert scale (below average (score < 2.5), average (score 2.5–3.4), above average (score > 3.4)).

The questionnaire presented to restaurant owners (Appendix A) was also developed based both on Rogers’ Diffusions of Innovation Theory and on the Perceptions of adopting an information Technology Innovation [22,25]. Additional questions regarding: (1) Complexity: what does the restaurant owner perceive as the major difficulties/limitations of the different menu labeling formats; (2) Testability: can the labels be easily and rapidly tested by the restaurant owners; and (3) Visibility: can the menu labeling format be used to add a distinctive value to the restaurant owners, were asked.

### 2.3. Menu Labeling Formats

Three menu labeling formats were selected from literature and implemented in the theory-based questionnaires: a simple health logo, star ranking, and calorie information. 

The selected simple health logo (Figure 1a) bears a strong resemblance to the at that time well-known Choices logo, which used to be visible on packaged foods in the Dutch supermarkets [26]. The recognizability of this logo ensures the logo to be easily understood by the (potential) restaurant guests. The ranking labeling format (Figure 1b) was selected because this format provides a quick and easy way to compare meals based on nutritional quality. Specifically, a star ranking was selected since it can be considered as the least confronting way of ranking, as opposed to, for example, a color ranking (from red to green) [27]. The calorie information labeling format (Figure 1c) was designed to merely provide information on the number of calories of the meals and was selected because this labeling format has been used in the US since 2010 [12,15]. Figure 1 shows how the different menu labeling formats may appear on the menu.

### 2.4. Statistical Analyses

The data from (potential) restaurant guests were analyzed using Jamovi version 1.6.21 for Windows. Boxplots combined with violin plots, means, and observations were presented as an indication of the variability and centers of the sample data. A χ^2^-test goodness of fit was used to determine the preference of the study population for a certain menu labeling format. A χ^2^-test of association was used to test for differences in label preference between subpopulations. Variables that were found to be related with label preference (age and ordinal level of education) were evaluated using a one-way analysis of variance (ANOVA) followed by pairwise comparisons of label-groups using the Tukey post hoc tests (equal variance) or Games–Howell post hoc test (unequal variance) and a Kruskal–Wallis (KW) test followed by Dwass–Steel–Critchlow–Fligner (DCSF) post hoc tests. Ordinal values assigned to the different educational levels are as follows: primary school = 1, lower vocational education = 2, preparatory secondary vocational education = 3, senior secondary vocational education = 4, senior general secondary and pre-university education = 5, higher professional education = 6, and academic education = 7. In some analyses, potential guests were split into younger (≤30 y) and older (>30 y) persons. 

Due to the limited number of participating restaurant owners (n = 41) no hypotheses were tested for this group.

## 3. Results

### 3.1. (Potential) Restaurant Guests

#### 3.1.1. Overall Characteristics

A total of 386 (potential) restaurant guest living in the Netherlands were included for analysis. Five provinces in the south and the west of the Netherlands (North Brabant (25%), South Holland (22%), Utrecht (15%), Gelderland (13%), and North Holland (9%)) were overrepresented. Concerning gender, the vast majority of the respondents (79%) were female. The study population was relatively young, 305 of the respondents were aged between 16 and 50 years (79%). Regarding education, the majority of the respondents (75%) followed or was following a higher professional education or an academic education. The majority of the respondents (55%) had a monthly income below €1500,-.

#### 3.1.2. Label Preference

The majority (42.7%) of (potential) restaurant guests indicated star ranking as the most appreciated menu labeling format (*p* < 0.001. There were 29.8% who indicated calorie information as their most appreciated labeling format and 27.5% who indicated the simple health logo as their most appreciated format. From all the participant characteristics determined, only age (*p* < 0.001) and educational level (*p* < 0.001) showed a significant relation with label preference. Regarding age, younger respondents (mean 31.9 year) were found to prefer ranking over the simple health logo (mean 40.1 year) and calorie information (mean 37.7 year), whereas older respondents rated the simple health logo as the preferred way to communicate health information of the different menu items (Table 1). Figure 2 shows the age distribution of the (potential) restaurant guests indicating to prefer either the simple health logo, ranking, or calorie information and illustrates the preference for ranking of the younger age group. Since the younger age group was highly educated, label preference could be confounded by educational level. DSCF pairwise comparisons after a KW test show that the preference of the younger respondents for ranking was still found when the group was split in potential guests with low (1–5) and high (6–7) educational level (highest *p* values were 0.002 and 0.057, respectively, for ranking vs. information).

The χ^2^-test goodness of fit also showed a relation (*p* < 0.001) between educational levels and label preference. The (ordinal) education level of (potential) restaurant guests preferring the simple health logo was lower than that of (potential) restaurant guests preferring ranking (*p* = 0.018;) or information (*p* = 0.026), no difference in education level was found between (potential) restaurant guests preferring ranking and information (*p* = 0.8) (Figure 3).

As the majority of respondents is relatively young and has a high educational level, the experimental design is not balanced enough to make statements about the relation between educational level and label preference. However, a KW test showed differences in ordinal levels of education for label preferences within the young (185, specified as ≤30 years old, *p* = 0.038), where higher educated young (potential) restaurant guests prefer information over ranking (*p* = 0.042) and over the simple health logo (*p* = 0.106). A similar test showed differences in older age groups (201, specified as >30 years old, *p* = 0.052) where the higher educated prefer ranking over the simple health logo (*p* = 0.036) (Figure 4). 

χ^2^-test goodness of fit showed no significant relation between label preference and health consciousness (*p* = 0.184) or other characteristics of the (potential) restaurant guests (gender (*p* = 0.184), monthly income (*p* = 0.125), frequency of visiting a restaurant (*p* = 0.107), and reason to visit a restaurant (*p* = 0.639) (Appendix A).

### 3.2. Restaurant Owners

A total of 41 restaurant owners was included for analysis. Included restaurants varied from small bistro restaurants to high quality Michelin star restaurants. Moreover, some restaurant chains were included. All included restaurants are situated in the ‘s-Hertogenbosch region. The majority of the questioned restaurant owners (46.3%) indicated (star) ranking as their preferred menu labeling format, with 36.6% indicating the simple health logo as their preferred menu labeling format and 14.6% indicating calorie information as their preferred format. One of the restaurant owners did not indicate a preference.

In addition to the obtained data on label preference, data on the perceived difficulty of implementing menu labeling was gathered. As shown in Table 2 the perceived difficulty shows a tendency towards easy (average score 2.8). 

Considering the different areas in which restaurant owners expect difficulties when implementing menu labeling, the time it takes to calculate the nutritional information of the dishes was found to be the major perceived barrier for implementing menu labeling. On a scale from one (no problems) to five (many problems) the time to calculate the nutritional information scored on average three. Explaining the meaning of the menu label to (potential) restaurant guests (average score 2.4), coming up with dishes that meet the criteria for the menu label (average score 2.6), and preparing dishes that meet the criteria for the menu label (average score 2.4) were scored as less difficult.

Next to exploring the perceived difficulty of implementing menu labeling, it was also examined whether restaurant owners considered menu labeling as a useful tool to positively distinguish them from other restaurants. As also shown in Table 2, the majority of restaurant owners do not expect menu labeling to positively distinguish them from other restaurants (average score 2.8).

## 4. Discussion

This research aimed to provide insight into which menu labeling format is most promising in supporting healthy eating behavior in Dutch restaurants by investigating the preference for three distinct menu label formats (simple health logo, (star) ranking, calorie information). Both (potential) guests and restaurant owners preferred ranking above a simple health logo or calorie information. When these results were studied in more detail, we saw that age and level of education influenced the preference of the guests. 

### 4.1. (Potential) Restaurant Guests

We expected that the simple health logo would be scored as most preferred, since at the time that the research was performed, Dutch consumers were are already familiar with this label as it was widely used in supermarkets. However, (potential) restaurant guests preferred the ranking format. This contradiction may have been caused by the fact that the Choices logo at the time, was increasingly being criticized by different stakeholders [26,29].

It was hypothesized that label preference would be related to age, educational level, and health consciousness. The results did indeed show a significant relation between age and label preference, with a preference for ranking among the younger age groups. Although the older age group (51–75 y) was somewhat smaller (19% of total participants), there was a clear shift in label preference. There are indications for a relation between the preference for a certain menu labeling format and the level of education, where lower educated (potential) restaurant guests tend to prefer the simple health logo, whereas labeling by either ranking or information was preferred by the higher educated guests. This may be explained by the fact these latter two types of labeling require certain interpretation by the (potential) restaurant guests, i.e., the number of calories in relation to total daily calorie requirement. Furthermore, previous studies showed that there is a positive relation between educational level and health consciousness [30,31]. A study performed by Ellison et al. [23] has shown that customers with high health consciousness chose ranking (a traffic-light-system) over simple calorie information. Since the majority of our participants were highly educated, the higher health consciousness of the study population could also account for the preference for ranking. Surprisingly, however, we did not observe a significant relation between label preference and health consciousness. This might be due to the limited differences in health consciousness between the groups as compared to the overall variability which is small.

### 4.2. Restaurant Owners

To the best of our knowledge this is the first study to investigate the preference of restaurant owners for a particular menu labeling format and their views on implementation. Results showed that the preference of restaurant owners for a certain format was similar to that of (potential) restaurant guests. However, it should be noted that since only 41 restaurant owners filled out the questionnaire, data from the restaurant owners could not be statistically analyzed. Nevertheless, it did give a first glimpse of the preference of this specific group. With respect to expected barriers and opportunities for implementing menu labeling, results showed that restaurant owners perceive the time it takes to calculate the nutritional information as the major difficulty. In addition, they see little value of menu labeling in positively distinguishing them from other restaurants. Although these results are contradictory to what we have found in an earlier small qualitative study where owners, chefs, and serving personnel of four restaurants were interviewed [32], they were consistent with previous research, which showed that restaurants are faced with many challenges when implementing menu labeling [6].

### 4.3. Limitations of the Study

This study has several limitations. First, the sample predominantly consisted of highly educated, relatively young, female respondents. In addition, the low income together with the fact that the study was performed by students of the HAS University of Applied Sciences suggests that students may be overrepresented in the sample. This hinders the generalizability of the study’s findings to lower educated, male, and older respondents. Since highly educated respondents may already be more aware of their health behaviors, our data might be influenced by the unrepresentative sample. Second, in this study participants were asked which menu labeling format they would prefer, based on a fictive menu which was presented in an online questionnaire. Despite the fact that the labels were based on actual nutritional data of these menu items and ranked in accordance with the Choices criteria, customers could fill out the questionnaire at different moments of the day and at different places. If customers were asked to fill out the questionnaire in a restaurant setting, their preference might differ since restaurant guests usually behave differently in real-life situations than in hypothetical ones, such as in an online questionnaire [8,33]. Third, one of the menu labeling formats, the simple health logo, was increasingly being criticized at the time the study was performed, and the logo was taken from the market in 2018. This criticism may have affected the results. Fourth, the small number of restaurant owners hinders statistical justification of the results. Still this study was one of the first to take into account the overlooked role of restaurant owners in establishing label preference. Fifth, we only examined how (potential) restaurant guests and owners like the different menu labeling formats; we did not examine the perception, understanding, and usage of this information [33], as it was beyond the scope of this quantitative study. 

### 4.4. Implications for the Future

To combat overweight and stimulate healthier choices in full-service restaurants, menu labeling can be a promising strategy. Menu labeling not only informs guests, but also potentially stimulates restaurant owners to critically evaluate their menu’s. However, besides menu labeling, other strategies should be considered. A randomized controlled trial performed by Velema et al. [34] in 30 worksite cafeterias found that a mix of social marketing based strategies, tailored to the target audience and applied coinciding with nudging, is effective in stimulating healthier choices. Likewise, a social marketing review performed by Carins and Rundle [35] recommended looking for other strategies rather than solely communication and advertising. Another recent review based on 183 studies found that interventions in restaurant settings that make use of the social norms are promising [36]. A study in two full-service restaurants demonstrated the effectiveness of healthier defaults on the menu in nudging guests to make the healthy choice [37]. This underlines the importance to not merely focus on menu labeling as a nudging strategy, but also on other strategies such as social marketing and behavioral economic strategies. 

Regarding menu labeling preference, a challenging task for further research is to determine the menu label preference of guest and owners of Dutch full-service restaurants in a real-life setting and with a larger and more balanced study population. In addition, more research on this topic needs to be undertaken to examine the preference for a simple health logo other than the Choices logo, to confirm that potential guests and owners prefer ranking over another well-known simple health logo. In 2021, the Nutri-Score will be introduced as a front-of-pack label to support consumers in making a healthy choice and to encourage manufacturers and retailers to reformulate their products. Despite the fact that the Nutri-Score is less simple as it requires more information compared to the Choices logo, we recommend that future studies focus on the Nutri-Score as health logo instead of the Choices logo. Since the Nutri-Score combines a health logo and ranking, we expect this to be an effective menu labeling format for the out-of-home sector. However, good alignment with dietary guidelines will be conditionally for the use of any label format, including Nutri-Score. Next to additional research on the preference for a certain menu labeling format with the ultimate goal to encourage restaurant guests to make the healthy choice, further research to the effect of menu labeling on restaurants’ behavior is necessary. 

## 5. Conclusions

In summary, this study is one of the first to take into account the preference of the uninvestigated group of restaurant owners in addition to the well-researched group of restaurant guests regarding menu labeling formats. The results of this study suggest that ranking could be used to indicate the healthy choice on restaurant menus as a promising way to improve healthy eating. Future research should try to determine the menu label preference of restaurant guests and owners in a real-life setting, with a larger and more balanced study population and with another simple health logo, e.g., the recently introduced Nutri-Score instead of the Choices logo used in this study. Moreover, including the effect of menu labeling on the healthiness of menu offerings is recommended. Finally, given that different strategies have proven to be effective in stimulating the healthier choice in full-service restaurants more research is required to further investigate the effect of menu labeling in combination with other strategies.

## Figures and Tables

**Figure 1 ijerph-18-12500-f001:**
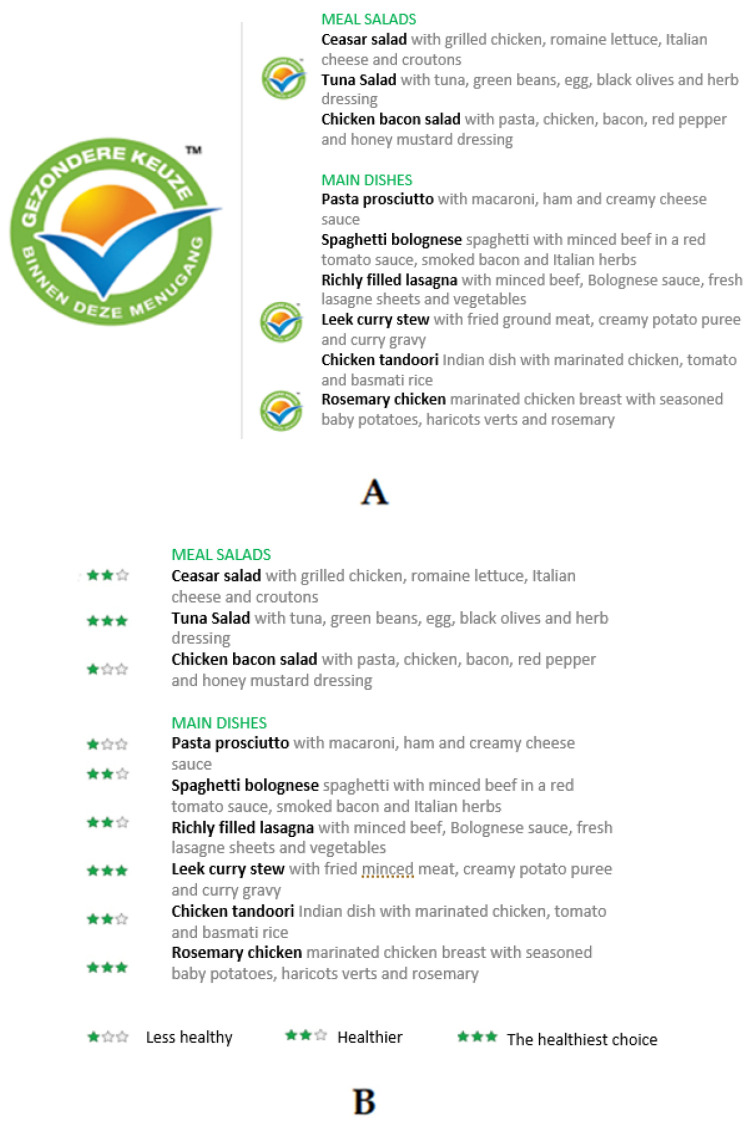
Fictive menu with implemented menu labeling formats: (**A**) simple health logo; (**B**) star ranking; (**C**) calorie information. A dish receives the simple health logo if it complies to all the choices criteria for main dishes. With star ranking the number of stars a dish receives is also established by the compliance to the Choices criteria for main dishes: compliance to all criteria = 3 stars; compliance to all minus max two criteria = 2 stars; compliance to less than two criteria = 1 star. With calorie information the number of calories is calculated based on NEVO data.

**Figure 2 ijerph-18-12500-f002:**
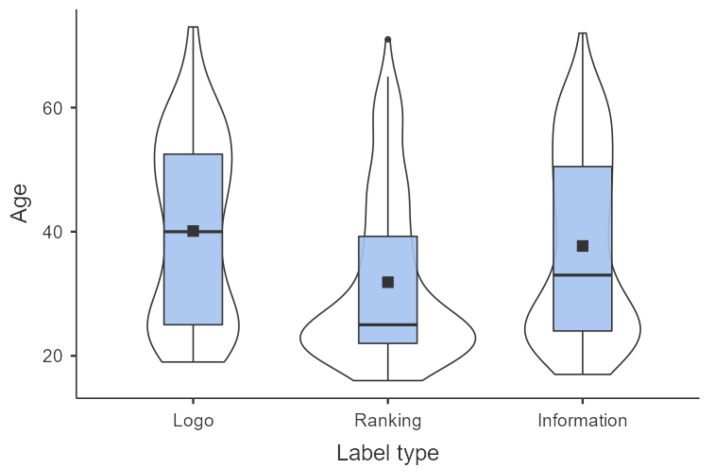
Box plot and violin plot of the age of the (potential) restaurant guests per menu label preference (logo, ranking, and information). The black square represents the average age per group (logo n = 107, ranking n = 164 and information n = 115). A box plot is a diagram with a box between the 25th and 75th percentile (IQR = interquartile range = P75–P25), a horizontal line at the median and a mean (square marker). The vertical lines (‘whiskers’) indicate the level of the highest or lowest observation within a distance of 1.5 IQR from the box. The violin plot is a smoothed density plot, where wide sections indicate a high density of observations [28].

**Figure 3 ijerph-18-12500-f003:**
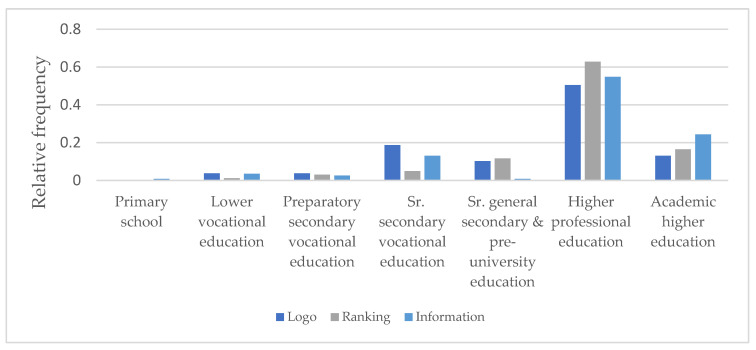
Relation of educational level of the (potential) restaurant guests with label preference (values are represented as relative frequencies). Educational level (*p* < 0.001; χ^2^-test goodness of fit) shows a relation with label preference.

**Figure 4 ijerph-18-12500-f004:**
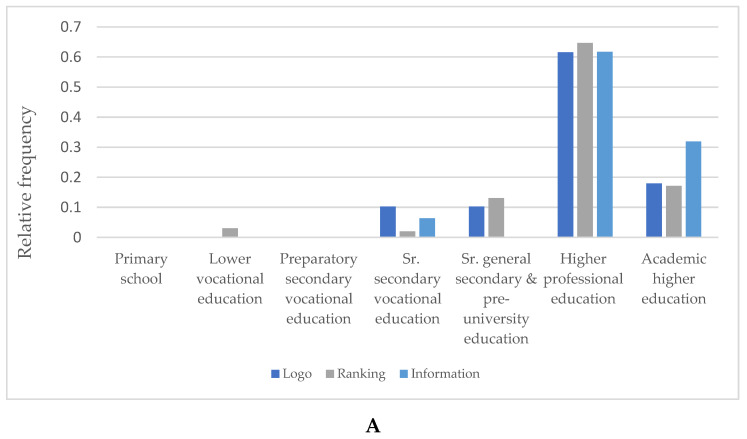
Relation of educational level of the (potential) restaurant guests ((**A**) ≤30 years old, (**B**) >30 years old) with label preference (values are represented as relative frequencies) corrected for age.

**Table 1 ijerph-18-12500-t001:** Mean age (ANOVA ^1^) and median age (KW ^2^) of groups expressing a certain menu label preference (logo, ranking, or information).

	Logo (n = 107)	Ranking (n = 164)	Information (n = 115)	*p*-Value
mean (SD)	40.1 (14.7) ^a^	31.9 (13.2) ^b^	37.7 (14.7) ^a^	<0.001 (ANOVA)
min–median–max	19–40–73 ^a^	16–25–71 ^b^	17–33–72 ^a^	<0.001 (KW)

Age (*p* < 0.001; KW) shows a relation with label preference. ^a, b^ Same letters (a, b) for group means and medians indicate no rejection of H_0_ (difference = 0) in post hoc tests (Games–Howell and DCSF, *p* < 0.001 except Games–Howell: logo vs. ranking *p* = 0.002). ^1^ Analysis of variance, ^2^ Kruskal–Wallis.

**Table 2 ijerph-18-12500-t002:** Extent to which restaurant owners (n 39) consider it complex to implement menu labeling on their menu (A), and extent to which restaurant owners (n 41) think implementing menu labeling positively distinguishes them from other restaurants (B). Numbers represent the frequency of restaurant owners who selected the respective degree of respectively complexity and distinctiveness.

	1	2	3	4	5	Average Likert Score
A. Perceived difficulty of implementing menu labeling * (5-point Likert Scale—1 = easy, 5 = difficult)	10%	33.3%	30.8%	20.5%	5.1%	2.8
B. Distinctive value of menu labeling (5-point Likert Scale 1 = no distinctive value, 5 = distinctive value)	7.3%	34.1%	34.1%	17.1%	7.3%	2.8

* Two restaurant owners did not answer this question.

## Data Availability

All data was anonymized and stored on a protected server only accessible by selected members of the research team. Anonymized data is available from the corresponding author upon request. Due to privacy and ethical restriction data is not publicly available.

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
