# Peer review of "What Is on the Menu?—A Quantitative Analysis on Label Format among (Potential) Restaurant Guests and Restaurant Owners"

_ijerph, 2021, doi:10.3390/ijerph182312500_

Round 1

Reviewer 1 Report

The subject matter and potential impact of these findings was very interesting and unique. The manuscript is well-written and clear and requires only a few small edits to some of the English.

Unfortunately, as the authors address, the large skew of respondents with regard to age and education level makes the results rather obvious that differences in attitudes about the menus is influenced by this.

At least in the journals with which I am familiar, there is more information given in the Methods section about the exact questions and scales and don't appear for the first time in the Results. This may differ by journal and just happens to be my preference.

In reference to the results again, the repeated reference to the (p-value and chi square test/goodness of fit) for several data seems unneeded, and with some simple sentence restructuring could be stated once for all of them.

Again, the writing clarity was very good, but the significance of the results, while they really could have been very insightful, is hindered by the skew of the participation demographics.  If gathering more data was an option, I would suggest it, but then the logo symbol used at the time of data collection is now irrelevant.  Another consideration that would have made this study more in keeping with its state objectives would have been to gather data on what kinds of menu choices the subjects would have been made (compared to a control group without the labels) to see if, in fact, the labels were impactful. Preference towards a label may not mean that it changes food choice.

Author Response

Dear reviewer,

First of all, we would like to thank you for the valuable and constructive comments and remarks on our manuscript entitled “What is on the Menu? - A Quantitative Analysis on Label Format among (Potential) Restaurant Guests and Restaurant Owners”. Herewith we would like to submit a revised version of the manuscript. Please see the attachment for our detailed response to your comments.

Sincerely,

Nadja Hanssen, MSc

Reviewer 2 Report

Nowadays, when we face a pandemic of obesity, menu labelling could be considered a tool to help consumers make informed choices. Since taste preference is an important factor influencing consumers' food choices, choosing the right way to inform consumers is important. Because of this, this paper could contribute to this endeavour, especially as it also presents managers' views on menu labelling.
Some minor changes should be made before accepting for publication. The changes are as follows:

- Ln 137- please provide a detailed explanation of the term "health consciousness" and the criteria. Also, the results section is missing the data on health consciousness are missing. These results should be added to manuscript not as supplementary files.  
- Please include the questionnaire as an appendix.
Ln 151- 170 - please explain the criteria used to award the single health logo and the star logo in more detail. What exactly does one star, two stars, three stars, etc. mean?
- In all figures, please remove the titles from the Excell illustration. Titles don't need to be written twice.

Author Response

(The authors gave the same response as above.)

Reviewer 3 Report

This is an overall well-written paper. However, I do have some minor concerns:

  • Abstract (line 15): You need to explain why restaurant owners are an overlooked group. In which regard?
  • Abstract: I am missing information about the sample size in the abstract.
  • There are some minor spelling mistakes which should be corrected.
  • You should also refer to the selection bias due to the sampling design in the methods section. It is not only an overrepresentation higher educated and younger people.
  • Did the study receive any ethics approval?
  • Figures 5 and 6 can be combined in one table.

Author Response

(The authors gave the same response as above.)

Round 2

Reviewer 1 Report

I am satisfied with the corrections made by the authors.